# The monthly evolution of precipitation and warm conveyor belts during the central southwest Asia wet season

Melissa Leah Breeden[1,2], Andrew Hoell[2], John Robert Albers[2], and Kimberly Slinski[3,4]

[1]Cooperative Institute for Research in Environmental Sciences, University of Colorado Boulder, Boulder, CO
[2]NOAA Physical Sciences Laboratory, Boulder, CO
[3]NASA Goddard Space Flight Center, Greenbelt, United States
[4]Earth System Science Interdisciplinary Center, University of Maryland, College Park, United States

*Correspondence to*: Melissa L. Breeden (melissa.breeden@noaa.gov)

**Abstract.** Understanding the nature of precipitation over central southwest Asia (CSWA), a data-sparse, semi-arid region, is important given its relation to agricultural productivity and the likelihood of hazards such as flooding. The present study considers how daily precipitation and local vertical motion – represented by warm conveyor belts (WCBs) – evolve from November to April over CSWA. We first compare several precipitation datasets, revealing the seasonality of daily precipitation
is consistent across estimates that incorporate satellite information, while total accumulation amounts differ substantially. A common feature across datasets is that the majority of precipitation occurs on the few days when area-averaged accumulation exceeds 4 mm, which are most frequent in February and March. The circulation pattern associated with heavy ($> 4$ mm day$^{-1}$) precipitation days evolves within the wet season from a southwest-northeast tilted couplet of circulation anomalies in January and February to a neutrally-tilted monopole pattern in April. El Niño conditions are associated with more heavy precipitation
days than La Niña conditions, with both enhanced WCB frequency and moisture transport observed during the former. An exception to this is found in January when precipitation, WCB frequency and moisture do not increase, despite a similar increase in surface cyclones to other months, suggesting precipitation changes cannot always be inferred from cyclone frequency changes. Nonetheless, our results generally support prior connections made between ENSO and seasonal-to-interannual precipitation anomalies and extend this connection to one between the slowly-evolving ENSO influence and
transient and local vertical motion represented by WCBs.

## 1 Introduction

Central southwest Asia (CSWA) includes several food-insecure countries – Afghanistan, Pakistan, Tajikistan and Uzbekistan (FAO, 2022) – that depend heavily on cold season precipitation, as the mountain snowpack and subsequent snowmelt runoff provide a vital source of water for agriculture (Thenkabail et al. 2004; Lotsch et al. 2005; FEWS NET 2018;
Qutbudin et al 2019; McNally et al. 2022). Afghanistan in particular receives the vast majority of its precipitation from November – April (Hoell et al. 2015a; Gerlitz et al. 2020), with substantial precipitation generated as flow is forced, often by westerly cyclonic disturbances, up the high topography of the Hindu Kush mountain range in the central and northeastern portions of the country (Mooley 1957; Sharma and Subramaniam, 1983; Lang and Barros 2003; Cannon et al. 2017; Hunt et

al. 2017). The area is also prone to flooding (Hagen and Teufert 2009) and landslides (Hunt and Dimri 2021), particularly during spring when rain-on-snow events are most common, with substantial impacts to lives and property. CSWA precipitation during the cold season has substantial interannual variability, making the region susceptible to drought, such as the devastating 1999-2001 event that occurred during persistent La Niña conditions (Barlow et al 2002). In part due to the crucial but unreliable role that precipitation plays in the politically and socioeconomically turbulent region, humanitarian food aid is provided to the country by multiple sources including the United States Agency for International Development (USAID), with regularly-issued weather and climate outlooks for Afghanistan provided by the Famine Early Warning Systems Network (FEWS NET; Funk et al. 2019; McNally et al. 2022). The lack of reliable in situ precipitation measurements over CSWA means that estimating precipitation relies on datasets incorporating satellite-derived quantities (e.g., Funk et al. 2015), and that determining which estimate is 'correct' is not necessarily well-posed. Still, when multiple estimates display agreement, the confidence in the precipitation estimate is enhanced.

Precipitation displays strong variability across timescales from minutes to months (e.g., Trenberth et al. 2017), with large-scale, slowly-evolving phenomena – including the seasonal cycle – modulating, over many regions, the frequency of short-duration, high-amplitude events. Hoell et al. 2015a found that months of anomalously low CSWA precipitation differed from pluvial periods by the presence of equivalent barotropic anticyclonic anomalies located just to the west of Afghanistan, with moisture fluxed southwestward on their eastern flank, also supporting anomalous drying. They demonstrated how different months within the cold season were associated with a variety of combinations of Indo-Pacific sea surface temperature (SST) patterns. It has been well-established that SST variations and the associated tropical convective heating response can modulate CSWA precipitation through the generation of an anomalous Rossby wave source and large-scale vertical motions (Barlow et al. 2002; Hoell et al 2012; Hoell et al. 2013; Hoell et al. 2015b; Hoell et al 2018a; Hoell et al. 2018b). In particular, the El Niño-Southern Oscillation (ENSO) is a strong modulator of southwest Asian climate, with La Niña associated with reduced moisture fluxes (Mariotti et al. 2007) and an anomalous upper-level anticyclone and downward vertical motion (Nazemosadet and Ghasemi 2004; Hoell et al. 2014a; Hoell et al. 2014b; Hoell et al. 2015a; Breeden et al. 2022), both conducive to reduced precipitation that can lead to drought (Barlow et al. 2002). Conversely, anomalous rising motion and enhanced precipitation are observed during El Niño conditions (Hoell et al. 2017). The North Atlantic Oscillation (NAO) may also influence precipitation over the Western Himalaya region, although recent studies suggest the precipitation impacts are located to the east of Afghanistan (Hunt et al. 2022), and that the NAO impact on the region may have weakened in recent decades (Yadav et al. 2009).

Past research has focused heavily on CSWA precipitation variability averaged over the November – April wet season, or from the perspective of monthly means, consistent with the interest on long-lived precipitation and soil moisture deficits associated with drought. While precipitation in this region is undoubtedly affected by low-frequency climate oscillations, such as ENSO or the seasonal cycle, precipitation ultimately varies on shorter timescales on the order of hours and days, with a small number of intense events contributing an outsize amount to the long-term mean (Pendergrass and Knutti 2018), as has been previously acknowledged to be the case in this region (Hoell et al. 2015a). The link between transient 'westerly

disturbances' and CSWA precipitation has been discussed in prior studies (Barlow et al. 2016; Hunt et al. 2017; Cannon et al. 2017; Hunt et al. 2021), including composite analysis of westerly disturbances year-round (Hunt et al. 2017). However, the month-to-month differences in the circulation patterns associated with substantial daily precipitation accumulation have not been considered in the present literature. A global study of warm conveyor belts (WCBs), the warm, moist, strongly ascending airstreams associated with extratropical cyclones (e.g., Green et al. 1966; Browning and Emanuel 1990; Wernli and Davies 1997; Madonna et al. 2014; Pfahl et al. 2014) found that a substantial portion of total annual precipitation over CSWA was associated with these features (Pfahl et al. 2014), though the seasonality and circulation patterns associated with WCBs in this region have not been examined in detail. As such, how daily precipitation and WCBs associated with CSWA precipitation evolve within the months that comprise the boreal cold season is the focus of the present analysis.

This study explores how daily precipitation, and the associated WCBs and circulation patterns, evolve within the months that comprise the CSWA wet season. Section 3.1 compares precipitation estimates from several datasets including satellite-derived, reanalysis, and gauge-based products and reveals how in all datasets, a small number of heavy precipitation days dominate total accumulation. Section 3.2 explores the seasonality of WCBs and precipitation, including a month-by-month comparison of heavy precipitation events and their attendant circulation. Finally, Section 3.3 considers the impact of ENSO on WCBs and precipitation, revealing that ENSO conditions modulate the frequency of WCBs and heavy precipitation days, which dominate monthly mean precipitation changes.

## 2. Data and Methodology

### 2.1 Data

Afghanistan is a data sparse region (Hoell et al. 2015a; Sun et al. 2018), so first we verify that the precipitation characteristics discussed are not a function of the precipitation dataset chosen, by comparing the statistics of daily mean precipitation from five different products: two reanalysis products, two remote-sensing and gauge-based products, and one solely gauge-based product (Table 1; Figs. 1–2). The longest common period for JRA55, CHIRPS, ERA5 and CPC precipitation is 1981–2020, while the shorter common period of 2001-2019 is also considered to include a comparison with IMERG. CHIRPS and IMERG are combined satellite-gauge products, although relatively few (between 0 and 55, depending on the year; CHC UCSB 2023) rain gauges are located within Afghanistan, meaning these precipitation estimates are largely satellite-derived. JRA55 and ERA5 precipitation are reanalysis products and are therefore estimates that are physically consistent with other variables in the model. Reanalysis precipitation estimates are the outcome of the assimilation of a range of observations, which may be spatially inhomogeneous over long periods, and model data used to fill in the spatial and temporal gaps. ERA5 and JRA55 precipitation are available at short latency (3–5 days for JRA55 and ERA5, respectively), making them amenable to generating near real-time forecasts. As such, knowing how these two reanalysis products compare to alternatively-derived precipitation estimates from CHIRPS and IMERG is considered valuable. Finally, we include the NOAA Climate Prediction Center (CPC) Unified Gauge precipitation dataset to consider the representation of precipitation from a purely gauge-based product. We also emphasize that given the sparsity of in-situ precipitation records over Afghanistan and that the number of stations in the country has historically varied markedly (CHC UCSB 2023), the goal of comparing the

datasets is to confirm that the aspects of precipitation considered are consistently observed by different products, and less to emphasize which product is the most 'correct' regarding exact precipitation amounts.

To consider the circulation associated with heavy precipitation events, daily mean ERA5 200- and 750- hPa streamfunction fields and 3-dimensional relative vorticity and potential temperature fields were accessed at 2.5x2.5° horizontal resolution and daily mean ERA5 vertically integrated zonal and meridional moisture fluxes ($IVT_x$, $IVT_y$, respectively; Eqns 1–2, units kg m$^{-1}$ s$^{-1}$) were accessed at .25x.25° horizontal resolution (Hersbach et al. 2020), where u and v denote the zonal and meridional wind (m s$^{-1}$), respectively, q indicates specific humidity (kg kg$^{-1}$), g is the gravitational acceleration of Earth (m s$^{-2}$), p represents pressure (Pa) and $p_s$ represents surface pressure (Pa).

$$IVT_x = \frac{1}{g} \int_{10}^{p_s} q\, u\, dp \qquad \text{Eqn 1}$$

$$IVT_y = \frac{1}{g} \int_{10}^{p_s} q\, v\, dp \qquad \text{Eqn 2}$$

To track strong upward vertical motion associated with extratropical cyclones, the warm conveyor belt (WCB) frequency dataset (Madonna et al. 2014; Sprenger et al. 2017) was used, which is based on ERA-Interim reanalysis. Events during the period 1981–2017 are considered to be consistent with the precipitation and WCB datasets (CHIRPS begins in 1981, and WCBs are tracked through 2017). In this dataset, WCBs are tracked by locating air parcel trajectories within the vicinity of an extratropical cyclone that originate in the lower troposphere and ascend at least 600 hPa within two days. As such, WCB occurrences highlight regions of strong ascent that are associated specifically with low-level cyclones. The frequency of 6-hourly timesteps indicating a WCB were averaged to daily means and is considered at 1x1° horizontal resolution. To partition months by ENSO phase, the Oceanic Niño Index (ONI), which is based on the Niño3.4 index (Trenberth 1997) that averages monthly mean SSTs in the region 5°N–5°S, 170°W–120°W and applies a three-month running mean, was acquired from the CPC (https://origin.cpc.ncep.noaa.gov/products/analysis_monitoring/ensostuff/ONI_v5.php). In section 3, El Niño (EN) conditions are defined when the ONI is at least a value of +0.5°, and La Niña (LN) conditions are defined when the ONI is less than or equal to -0.5°.

### 2.2 Precipitation Duration and Intensity

We consider the probability of daily precipitation, area-averaged precipitation over Afghanistan (box in Fig. 1), as a function of duration and daily accumulation. Two-dimensional probability distributions are determined by tracking the number of events of varying duration and threshold, in increments of days and 0.25 mm, respectively. First, all dates in a month above a selected threshold are extracted, and the length of each string of consecutive dates exceeding that threshold is counted and binned, with a requirement of one day of separation between distinct events. The number of events longer than each bin's duration are also summed and added to each duration's bin, to create cumulative distributions of event duration. Finally, each bin's event number is divided by the sum of all events in that month to attain the normalized probability of a precipitation event that is at least the value of that bin's duration and daily accumulation.

### 3. Results

### 3.1 Daily Precipitation Characteristics

Afghanistan's semi-arid climate receives most of its precipitation during the boreal cold season from November – April, during which time daily precipitation variance is generally comparable across IMERG, CHIRPS, JRA-55 and ERA5 datasets, while the gauge-based CPC dataset differs due to the limited number of stations in the region (Fig. 1). Daily statistics of area-averaged precipitation over Afghanistan (box in Fig. 1) also display good agreement among the various datasets considered, with the exception of the CPC estimate, and with better agreement between the standard deviation and 95[th] percentile values than the medians (Fig. 2), reflecting the non-Gaussian nature of precipitation (e.g., Papalexiou and Koutsoyiannis 2013). The higher precipitation totals over the Karakoram and Himalayan Mountain ranges to the east of the boxed region in Fig.1 have been excluded from the area-averaging due to differences in the annual cycle of precipitation related to the summer Indian monsoon, which does not routinely reach Afghanistan as it does Pakistan and northern India (Funk et al. 2015; Hunt et al. 2017). JRA55 displays the highest median and standard deviation, and CPC's gauge-based estimates the lowest. A common element among all datasets is that precipitation median and standard deviation peak at different times within the wet season. The standard deviation increases from October to February and decreases thereafter, while median precipitation does not increase notably until January, and remains elevated through April. This means that while accumulation is greatest in February, March and April, the dynamics and circulation patterns associated with precipitation in February, when the standard deviation and extreme values peak, is likely different than the nature of precipitation in April, when median precipitation is still high, but variance has decreased. These differences in the circulation and characteristics of precipitation within the peak precipitation months from January to April is the focus of subsequent analysis in section 3.2.

Binning area-averaged total precipitation over Afghanistan by each day's accumulation value indicates that the greatest portion of precipitation occurs on the few days with daily accumulation > 4 mm (Fig. 3), which we will refer to as heavy precipitation days. While there are some differences between datasets concerning total precipitation amount, with CHIRPS showing lower accumulations and JRA55 the highest, the month-to-month evolution of the different accumulation groups is consistent among the three datasets for their common period, 1981–2020. In all datasets, each month is likely to expect about 1–6 days on average with heavy precipitation, a reflection of the intermittent and intense nature of precipitation (e.g., Pendergrass and Knutti 2018), with the intensity likely related in part to orographic enhancement of vertical motion. March displays the greatest accumulation, followed by February, with heavy precipitation days dominating accumulation in both months. In April, the number heavy precipitation days decreases, while the contributions from values < 4 mm are elevated compared to earlier in the season, accounting for the reduction in standard deviation and 95[th] percentile precipitation values observed in Figure 2, while the median is similar to February and March. April is one of the top months for total precipitation, with totals similar or higher to the amount that occurs in January, depending on the dataset. Few heavy precipitation events last longer than a couple of days, with the longest events occurring in March and February (Fig. S1), consistent with the greatest number of heavy precipitation events observed during these months (Fig. 3). No heavy precipitation events are observed to last longer than a week in all datasets, with one-day events accounting for the majority, reflecting the intermittent nature of heavy precipitation in this area.

Considering the joint probability of precipitation duration and amplitude further demonstrates how precipitation events evolve from winter into spring, with more short, extreme events during the former and longer, weak events during the latter (Fig. 4; Fig. S2). The warm colors indicate the least likely but high accumulation precipitation events, while the highest probability contour in blue indicates more likely, lower accumulation events of less consequence. The constant liquid equivalent values shown in dashed black lines indicate where the accumulation of events with different duration and intensity is equal. In October, precipitation is generally scarce, with only short, low-amplitude events observed and probability values clustered around low accumulations in the bottom-left corner (Fig. 4a). In January and April, precipitation becomes more intense and more persistent, with increases in the likelihood of high accumulation events relative to October (Fig. 4b–c). Comparing events in January and April, which have comparable mean precipitation, there are clear differences in how precipitation occurs, which was suggested by the differences in total accumulation contributed from various daily accumulation values (Fig. 3). Specifically, January has a greater occurrence of short, high-amplitude events that exceed 12 mm accumulation (c.f., top-left quadrant of Fig. 4b vs c) while April has a greater likelihood of long, low-amplitude events that can produce similar accumulation totals (c.f., bottom-right quadrant of Fig. 4b vs c). The next section will show that, in addition to these differences in precipitation duration and amplitude, the circulation associated with heavy precipitation days differs between January and April as well.

### 3.2 Warm Conveyor Belts and CSWA Precipitation

While seasonal and monthly mean precipitation over CSWA has been related to enhanced moisture transport in past studies (Mariotti 2007; Hoell et al. 2015a), the transient features that evolve on *daily* timescales and are related to precipitation have not been extensively considered, particularly regarding their month-to-month differences. The seasonality of the WCB frequency indicates that strong ascent associated with extratropical cyclones peaks from February to April, consistent with the timing of peak precipitation (Figs. 5-6). This result is consistent with Pfahl et al. 2014, who found annual CSWA precipitation was heavily related to WCBs – with roughly 50% of total annual precipitation, and an even higher fraction of extreme precipitation, associated with WCBs over CSWA – and shows that this correspondence is greatest in late winter and early spring. We acknowledge that some precipitation might not be associated with WCBs in this region, though given the several requirements to identify WCBs, one can imagine that some ascending trajectories associated with extratropical cyclones come close to, but do not quite meet, the WCB criteria. WCBs may therefore be considered as a reflection of maximum ascent associated with extratropical cyclones, with the understanding that not every trajectory experiencing ascent and condensation is captured by this one definition.

WCBs are largely absent over the region from September – November, increasing in frequency through March and thereafter decreasing to very rare occurrences by June. The region affected by WCBs is oriented in a southwest-northeast manner across the middle east and southwest Asia (Fig. 6), on the cyclonic shear side of the upper-level jet. However, while the jet is strongest in January and February, WCB frequency peaks in March and remains strong in April as well, when the jet has weakened substantially. This suggests that the upper-level storm track, which peaks when the jet is strongest in January-February (Barlow et al. 2016), is not the only factor influencing the frequency of WCBs in this area. A local maximum in

WCB frequency over Pakistan develops in March and persists through May, which is not observed in December-February, also reflecting how WCBs change during the progression from winter to spring. Given the WCB definition, requiring strong ascent within two days near a sea level pressure minimum, it is possible that cyclonic disturbances with different characteristics develop at various parts of the year, in the different mean states represented by the jet. This is explored next by comparing the circulation differences between days with heavy or negligible precipitation and how these differences evolve during the months of peak precipitation.

The composite difference between heavy ($> 4$ mm) and light ($< 0.04$ mm) precipitation days in each month confirms that, as the mean state evolves from the heart of winter into spring, the transient features associated with heavy precipitation change as well (Figs. 7–8). We note that $> 4$ mm events during the wet season display similar frequency and duration, even during the months January and April (Fig. 4) which displayed large differences in duration and frequency characteristics at lower accumulation thresholds. In January and February, a southwest-northeast oriented, low-level trough-ridge couplet are observed, with strong moisture transport and ascent located in-between the trough and ridge (Fig. 7a,b), similar to the composite structure of westerly disturbances examined by Hunt et al. 2017, which were found to be most frequent in January and February. This tilt combined with the cyclonic shear of the jet (Fig. 6) is conducive to barotropic energy extraction, which can support the amplification of baroclinic waves (Mak and Cai 1989), and thus could be a mechanism for development in this region. The positive tilt can also be reflective of anticyclonic wavebreaking and deep stratospheric intrusions, which have been linked to precipitation over the Middle East (de Vries et al. 2018) and could also be a factor for CSWA precipitation at this time of year. A similar pattern is observed at upper levels in the streamfunction and vorticity differences, though with an additional cyclonic difference located over southeast Asia, while a region of enhanced WCB frequency is collocated with the strong IVT over Afghanistan (Fig. 7e,f, 8a,b). In March, only a single 700-hPa cyclonic anomaly is observed, while the upper-level circulation involves a different pattern than January and February, with the hint of an anticyclonic anomaly located over northeast Africa that strengthens in April (Fig. 7g,h, 8c). The positive tilt to the anomalies observed in JFM is not present in April when the upper-level cyclonic (ie, positive) relative vorticity feature is strongest and dominates the streamfunction field, overall indicating a shift in the circulation associated with precipitation as winter transitions to spring that was similarly noted in Hoell et al. 2015a.

Vertical cross sections of the relative vorticity difference between wet and dry days, taken through the main dipole comprised of the positive vorticity difference located over Iran and Turkmenistan and negative difference located over northern India, Pakistan and southern China, indicates that the upper and lower vorticity fields are vertically collocated during all months, reflecting an equivalent barotropic structure (Fig. 8e-h). In-between these two vorticity anomalies is strong forcing for ascent (Martin 2006 and references therein) and thus the location of WCB formation and precipitation. The equivalent barotropic structure contrasts the vertical variation of the vorticity differences observed farther to the east over southern China indicative of a baroclinic structure. The upper-level cyclonic vorticity anomaly that is key in driving ascent and forming WCBs over Afghanistan is strongest between 350-200 hPa in all months and extends lower into the troposphere in January and February than March and April, despite strengthening in amplitude during the latter months. Conversely, the downstream

negative, anticyclonic vorticity difference remains of similar strength in all months. Considering how the streamfunction fields in Fig. 7 are the aggregate effect of all of the features in the vorticity field, it becomes clear why the cyclonic streamfunction differences are also stronger in April than in January. It also appears that the circulation associated with the two cyclonic vorticity features overwhelms the anticyclonic vorticity feature in-between (Fig. 8a-d), as the streamfunction fields mainly show the cyclonic differences (Fig. 7e-h). Composite isentropes on wet days tilt downwards towards the surface moving from the cyclonic to anticyclonic vorticity differences, reflecting warmer temperatures beneath the anticyclonic feature and cooler temperatures beneath the cyclonic feature.

Coinciding with the circulation changes, WCB frequencies also shift to peaking over Afghanistan in January and February to over Pakistan in March and April (Fig. 6). April is also characterized by locations of negative WCB frequencies to the west of the cyclonic anomaly, indicating a coincident suppression of precipitation in areas beyond Afghanistan associated with this pattern that is not present in other months. We note that March is in the middle of the winter-to-spring transition, which likely means that early in the month, precipitation occurs due to patterns that resemble January-February, while later in the month the pattern resembles that in April, meaning the overall March pattern is a blend with only a common element of a cyclone to the west. Future work could break down each month's heavy precipitation days into groups with similar patterns to better understand the flavors of precipitation that are observed within each month, which could be particularly illuminating during transition months such as March.

### 3.3 Modulation by El Niño and La Niña conditions

As previously mentioned, one of the most predictable signals altering precipitation over Afghanistan, and more broadly CSWA, is ENSO (Barlow et al. 2002; Barlow et al. 2016; Hoell et al 2020; Breeden et al. 2022). To confirm that ENSO differences are robust across multiple precipitation estimates, we first consider monthly difference in CSWA precipitation using ENSO events that occurred between 1981-2017 (Table S1). Over this period, EN conditions are most often associated with enhanced precipitation compared to LN conditions, particularly in March, when most significant increase in WCB frequency is also observed (Fig. 9). The precipitation differences between the two phases are consistent among the CHIRPS, ERA5 and JRA55 datasets, except for in April when CHIRPS indicates no significant change in precipitation, while ERA5, JRA55 – and WCBs – all display significant differences. In January, uniquely, no change in WCB frequency or precipitation is observed in any of the three precipitation datasets, which may initially be perplexing but is consistent with the lack of significant tropical Pacific SST anomalies associated with monthly CSWA precipitation anomalies in January (Hoell et al. 2015a). However, the precipitation dataset used by Hoell et al. 2015a was a reanalysis-based estimate that is adjusted using a blend of observational datasets, including satellite information, suggesting that the similarity between our and their results may be due to similar characteristics in the precipitation dataset, and may not serve as a fully independent verification. Despite this caveat in January, the generally observed increased precipitation during EN conditions is consistent with past studies considering seasonal means (Mariotti et al. 2007; Hoell et al. 2012; Barlow et al. 2016; Hoell et al. 2018a).

Higher mean precipitation during EN months is related to the increased occurrence of heavy precipitation days (Fig. 10), which increases from an average of 4.1 days in LN March to 5.9 days in EN March, and from 2.5 to 4.3 days in April LN and EN, respectively, for ERA5 precipitation. As such, only a few days of heavy precipitation are strongly influencing monthly-mean and seasonal-mean shifts in hydroclimate over the region, consistent with past precipitation studies highlighting this disproportionate influence (Pendergrass and Knutti 2018). Furthermore, while there are two fewer EN March months than LN March months (9 versus 11), more precipitation accumulated during the former (Fig. 10a-b), underscoring the disproportionate effect that short-lived, heavy precipitation during El Niño conditions has on CSWA hydroclimate. Finally, to explore whether WCBs are more intense or more frequent during EN months, we compare the median and 95[th] percentile values of WCB frequency for EN/LN conditions for the months February-March-April (Fig. S3). The median WCB frequencies are statistically significantly different, while the 95[th] percentile values are indistinguishable, implying that the strongest WCBs are not changing in frequency as much as WCBs of average intensity.

To better understand the month-to-month variations in precipitation during EN and LN conditions, we consider the composite IVT and WCB frequency in January, when precipitation and WCBs display no change between EN/LN conditions, and in March, when both display their strongest differences (Fig. 11). In January, WCBs are overall rare in the region (Fig. 6; Fig. 11a,b), with only small increases in WCB frequency observed during EN months, to the north of Afghanistan that are collocated with a positive precipitable water anomaly and enhanced northeastward IVT (Fig. 11c). Conversely, in March, WCBs are more frequent over a large area including Afghanistan, Pakistan, and northwestern China during EN compared to LN months (Fig. 11d-f). Positive differences in precipitable water between EN and LN months are observed locally and, to a larger extent, over the Arabian Peninsula, collocated with stronger southwesterly/westerly flow and enhanced IVT over the Arabian Peninsula, which is also the region where WCB trajectories begin their ascent (Madonna et al. 2014; Joos et al. 2023). The pattern of enhanced IVT is similar to the regression of IVT onto Niño3.4 in DJF (Mariotti et al. 2007). More abundant precipitable water during EN months could reflect an enhancement of the low-level moisture in the inflow region of the WCBs, conducive to enhanced diabatic heating and a subsequent strengthening of the cyclone and vertical motion (Madonna et al. 2014; Schäfler and Harnisch 2014; Joos et al. 2023), supporting more frequent WCB development. As such, during EN conditions, both vertical motion from synoptic systems *and* moisture availability from lower latitudes are strengthened, increasing precipitation, except in January when neither change. On heavy precipitation days during March EN months, a stronger cyclonic anomaly and higher WCB frequencies are observed than on March LN days (Fig. S4), consistent with enhanced ascent and precipitation during the former. WCBs and IVT also increase notably in February and April, in a similar manner to March, though WCB changes in in February are less uniform and mainly located to the south over Pakistan (Fig. S5). Overall, the exact timing of the biggest ENSO influence on CSWA precipitation appears to be in March and April, rather than earlier in the winter such as January, when the climatological paucity of WCBs seems to limit the influence on precipitation.

To investigate how surface cyclones change during EN versus LN conditions, monthly mean Eady growth rate (Eqn S1) and cyclone frequency datasets were attained from the same ETH ERA-Interim Lagrangian climatology database used to

identify WCBs ([http://www.eraiclim.ethz.ch/](http://www.eraiclim.ethz.ch/); Wernli et al. 2010). The Eady growth rate assesses the conditions for baroclinic instability, a key mechanism for cyclone growth (e.g., Eady 1949). Consistent with the increases in WCBs and precipitation that are observed in most months, both the frequency of cyclones and the Eady growth rate increase during EN versus LN conditions in a large region to the north and west of Afghanistan during peak wet season months from January – April (Fig. 12). The increase in these two quantities during EN months may indicate that the large-scale response to anomalous tropical heating results in an increase in the mid-level vertical wind shear, or a reduction in static stability, with either change potentially enhancing cyclone formation and increasing WCB frequency. However, unlike the month-to-month differences in precipitation and WCBs, particularly the minimal changes in January compared to other months, the cyclone frequency increases in EN January are similar to the increases observed in February, March and April, while the Eady growth rate difference is similar in its spatial pattern but weaker in amplitude. These differences suggest WCB and precipitation changes cannot always be inferred from cyclone frequency changes in this area, perhaps due to changes in the cyclone structure that aren't reflected as frequency changes (Joos et al. 2023). A similar discrepancy between a decreasing precipitation trend, despite no trend in westerly disturbance frequency, was found over the Western Himalaya region by Nischal et al. 2022 and was also noted from a global perspective by Joos et al. 2023. However, revisiting these month-to-month precipitation and WCB differences with a larger sample size using climate model ensembles, for instance, would be advantageous.

## 4    Discussion and Conclusions

This study compares the evolution of daily CSWA precipitation characteristics, with a focus on Afghanistan, to complement the consideration given to monthly and seasonal mean precipitation that have been the primary focus of many past studies. Precipitation estimates that incorporate satellite information display generally good agreement during the wet season, while the gauge-based CPC dataset suggests a lower estimate that differs substantially, given the sparse network of gauges in the area. Precipitation mainly occurs during short-lived, high-amplitude events during the cold season, with the patterns associated with precipitation evolving from a dipole-type pattern, indicated by a couplet of streamfunction anomalies, in January and February into a single cyclonic monopole pattern by April. The dipole pattern observed in January and February is similar to the 'central Asian weather type' identified by Gerlitz et al. 2018, who used k-means clustering over a large domain over central Asia to isolate different patterns of atmospheric variability during boreal winter (thus excluding April). The 'central Asia' pattern was the least frequent of the eight identified, consistent with the infrequency of precipitation in this region relative to others; this pattern also displayed the strongest precipitation anomalies over Afghanistan of all considered, consistent with the present analysis. Peak WCB frequency from February – April occurs later than the peak strength of the upper-level jet and storm track in DJF, suggesting that, in this region, upper-level dynamical forcing is not the only factor modulating the frequency of WCBs (Fig. 6). In addition to upper-level vertical motion forcing, it is possible that local moisture availability influences the observed precipitation seasonality, and indeed precipitable water reaches a minimum over Afghanistan in January (not shown). However, disentangling the ambient moisture supply from the moisture produced by cyclones and WCBs requires some consideration. As such, understanding the lifecycle of cyclones and exploring changes in

relevant factors, such as static stability, and the evolution of low-level moisture availability before and after cyclones pass through the area, could be a future avenue of research.

The intermittent nature of precipitation has been highlighted from a global perspective by Pendergrass and Knutti (2018), who also emphasized the disproportionate impact that only a handful of precipitation events has on the seasonal and
annual mean distribution. This uneven impact is reflected here for CSWA and highlights the region's susceptibility to drought, such as the one during the 1999-2001 La Niña (Barlow et al. 2002), when only 31 days with an accumulation exceeding 4 mm were observed in three years. This is roughly half of the long-term average of about 20 days per year with the same accumulation (Fig. 3), meaning that a leading impact of ENSO is to modulate the number of heavy precipitation days observed (Fig. 10). WCBs evolve on a similar temporal scale as precipitation by definition, and through their link to extratropical
cyclones (Madonna et al. 2014), are intuitively related to strong precipitation events. Showing that their frequency over CSWA is modulated by ENSO phase further clarifies the link between the low-frequency circulation changes that ENSO produces, and the transient, short-lived nature of precipitation in this region. The lack of CSWA precipitation sensitivity to tropical SSTs in January that was found by Hoell et al. 2015a is consistent with the similarity in WCBs between EN and LN conditions observed in January in this study. Why January in particular lacks such a link to tropical SSTs and ENSO, which is apparent
in the other cold-season months, remains to be determined and merits further investigation. Future work could decompose precipitation ENSO-related precipitation changes, or changes related to other teleconnection patterns such as the North Atlantic Oscillation, into intensity and frequency contributions following Catto et al. 2012 and Hauser et al. 2020.

Finally, we note that the choice of the Niño3.4 index to capture ENSO variability is motivated by the wide use and public availability of this index, and the fact that central Pacific ENSO events in particular appear to have an effect on southwest
Asian climate (Hoell et al. 2018a). It is true, however, that CSWA may be sensitive to certain characteristics of each ENSO event, such as the strength of the associated east-west SST gradient (Hoell et al. 2013), which is not necessarily captured by Niño3.4. Furthermore, there are alternative methods to capture ENSO variability than relying on indices of SST anomalies, such as linear inverse modeling, that have proven useful in capturing the full scope of ENSO's influence (Penland and Sardeshmukh 1995) and its impact on regional climate (Albers et al. 2021; Albers et al. 2022; Breeden et al. 2022). Future
work could make use of these dynamically-based techniques and revisit the question of ENSO's influence on central southwest Asian precipitation. Experiments generating larger samples of ENSO events in each month would also be useful to revisit the month-to-month variations in ENSO's influence found in this study. Finally, case studies of precipitation events during EN and LN conditions could elucidate the potential interactions between the ambient low-level moisture, WCB development and precipitation during different months within the wet season.


**Data Availability**

The JRA-55 Reanalysis data used in this study are freely available at https://doi.org/10.5065/D6HH6H41 (Japan Meteorological Agency, 2013), and CHIRPS precipitation is freely available at https://data.chc.ucsb.edu/products/CHIRPS-2.

0/global_daily/netcdf/p25/ (Climate Hazards Center, 2021). CPC Global Unified Gauge-Based Analysis of Daily

Precipitation data was provided by the NOAA PSL, Boulder, Colorado, USA, from their website at https://psl.noaa.gov. ERA5 Reanalysis data is freely available online: https://cds.climate.copernicus.eu/cdsapp#!/dataset/reanalysis-era5-single-levels?tab=overview. IMERG data is freely available online: https://gpm.nasa.gov/data/directory.

**Author Contributions:** MLB acquired data, wrote code for calculations, produced all figures, and wrote the manuscript. AH

and JRA offered guidance on relevant topics and datasets and provided input on figures and edits to the manuscript. KS provided input on figures and edits to the manuscript.

**Competing interests:** The contact author has declared that none of the authors has any competing interests.

**Acknowledgements:** The authors gratefully acknowledge two anonymous reviewers for their suggestions and comments, Dr. Michael Sprenger for generating the WCB dataset and Dr. Benjamin Moore for locally providing 6-hourly WCB dataset, and financial support from the Famine Early Warning Systems Network and United States Agency for International Development.

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

| Dataset | Period Considered | Horizontal Resolution | Temporal Resolution |
|---------|-------------------|-----------------------|---------------------|
| CHIRPS | 1981-2020 | 0.25x0.25° | Daily |
| IMERG | 2001-2019 | 0.25x0.25° | Daily |
| JRA55 | 1981-2020 | 0.56x0.56° | Daily |

| ERA5 | 1981-2020 | 0.25x0.25° | Daily |
|------|-----------|------------|-------|
| CPC | 1981-2020 | 0.5x0.5° | Daily |

**Table 1:** Summary of precipitation data used in this study: Japanese Meteorological Agency 55-year Reanalysis (JRA-55; Kobayashi et al. 2015); Climate Hazards InfraRed Precipitation with Stations (CHIRPS; Funk et al. 2015); European Centre for Medium-Range Forecasting (ECMWF) Reanalysis v5 (ERA5; Hersbach et al. 2020); Integrated Multi-satellitE Retrievals for GPM version 06 (IMERG; Huffman et al. 2019); and the NOAA Climate Prediction Center (CPC) United Gauge Product (Chen et al. 2008). Note that the 'temporal resolution' and 'data period used' columns indicate the time periods used in this study, and that each product may also be available at higher temporal resolution (ie, IMERG) or spatial resolution (ie, CHIRPS).

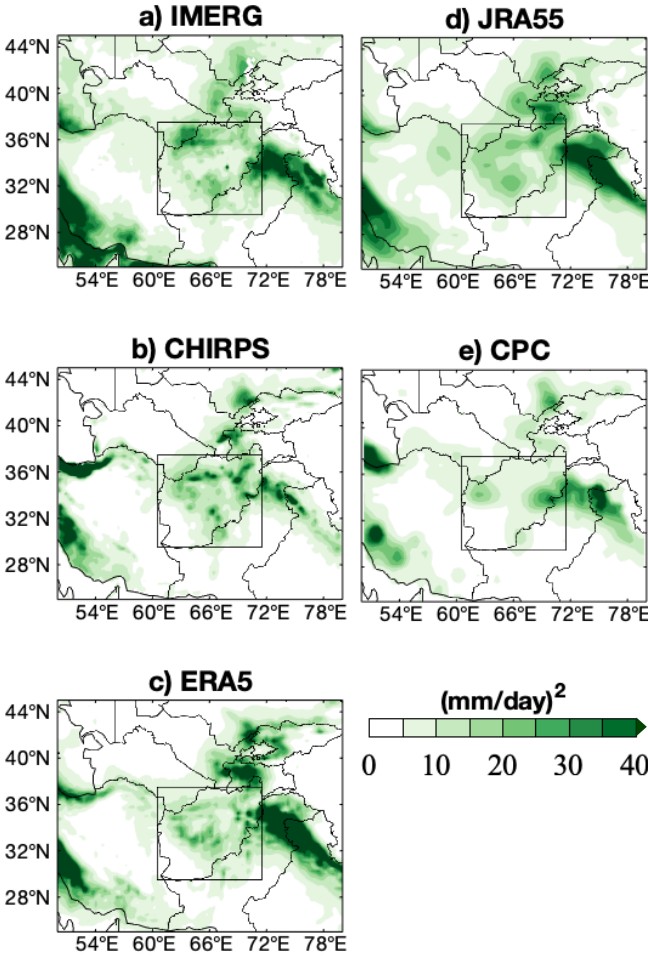


**Figure 1: November - April 2001-2019 precipitation variance for a) IMERG b) CHIRPS c) ERA5, d) JRA55 and e) CPC datasets. The black square outlines the domain where daily precipitation is averaged (29.5-37.5°N, 60.5-71.5°E). See Table 1 for details about each product.**

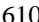

**Figure 2: Monthly evolution of daily mean precipitation median, standard deviation, and 95th percentile for CHIRPS, ERA5, IMERG, JRA55 and CPC datasets, for their common periods a) – c) 1981-2020 and d) – f) 2001-2019 averaged over Afghanistan (box in Figure 1). IMERG is also shown for the 2001-2019 periods.**

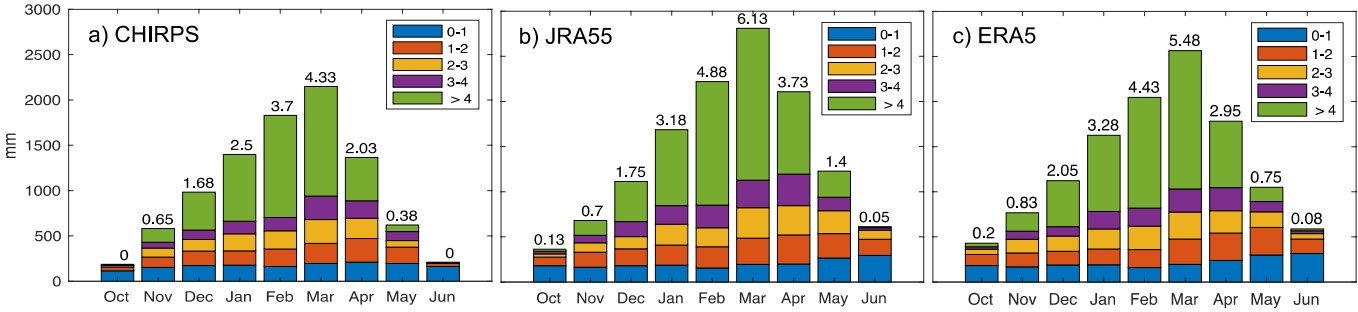


**Figure 3: Cumulative area-averaged (see box in Fig. 1) a) CHIRPS, b) JRA55 and c) ERA5 precipitation, binned by daily rate, summed over the years 1981-2020. The numbers at the top of each column indicate the average number of days per month with accumulation > 4 mm.**



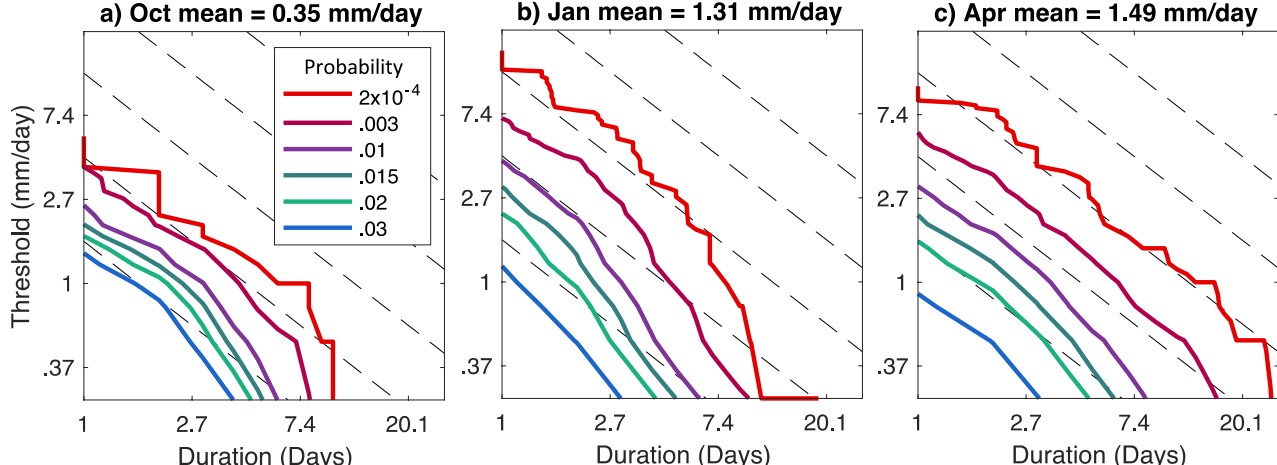

**Figure 4: The thick solid contours show the joint probability of daily ERA5 precipitation intensity and duration for a) October b) January and c) April, 1981-2020. The black dashed contours indicate lines of constant liquid equivalent for the values: 1.65, 4.48, 12.2, 33.1, and 90 mm (starting in the lower-left corner to upper-right).**



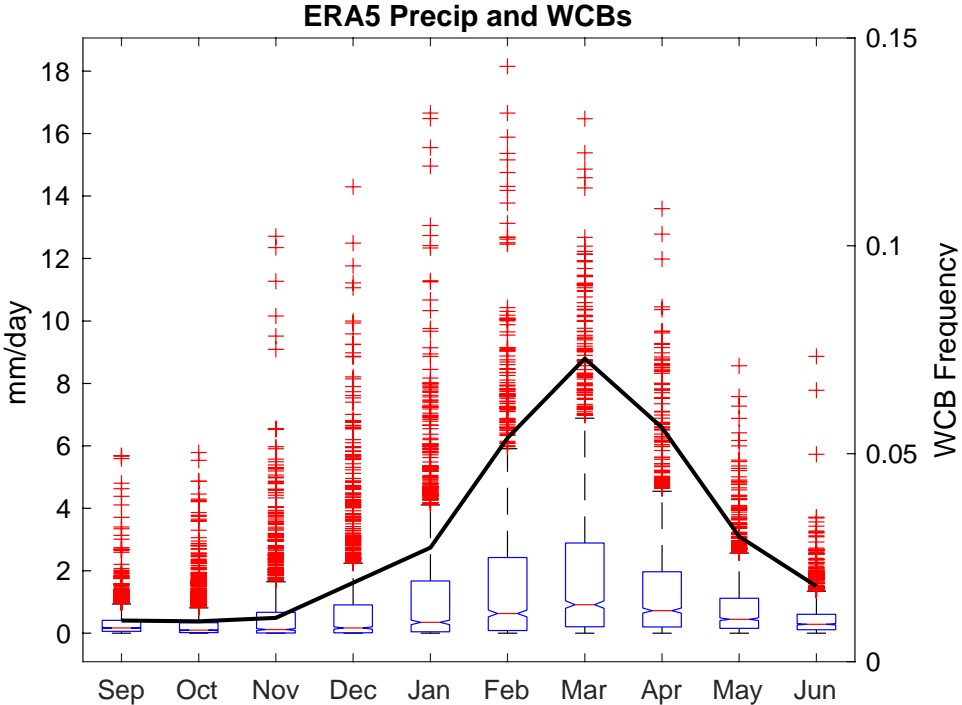


**Figure 5: Statistical distribution of ERA5 precipitation for September – June 1981-2020 (blue boxes and red '+' signs), and mean WCB frequency 1979-2017 (thick black line; units events per 6-hourly time step), both fields averaged over Afghanistan (box in Fig. 1). For precipitation, the interquartile range for each month is shown in the blue boxes, with the median indicated by the red line. Outliers are shown in the red '+' signs.**


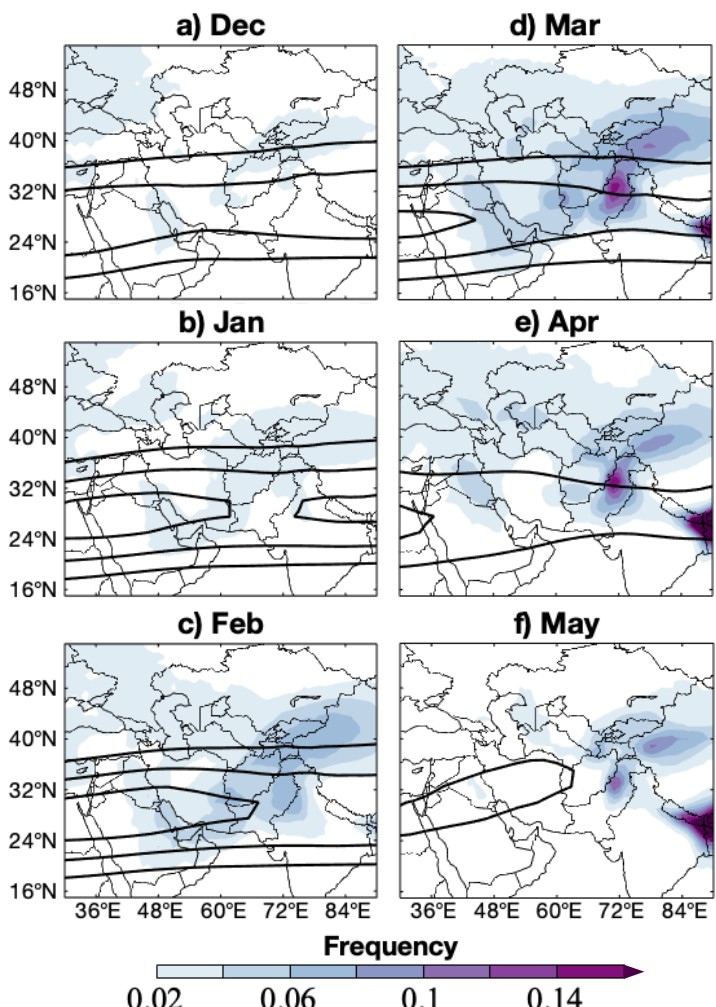

**Figure 6: The color shading shows 1981-2017 monthly mean WCB frequency in units of events per 6-hourly time step. The black contours show monthly mean 200-hPa zonal wind, contoured at 30 m s⁻¹ every 10 m s⁻¹.**

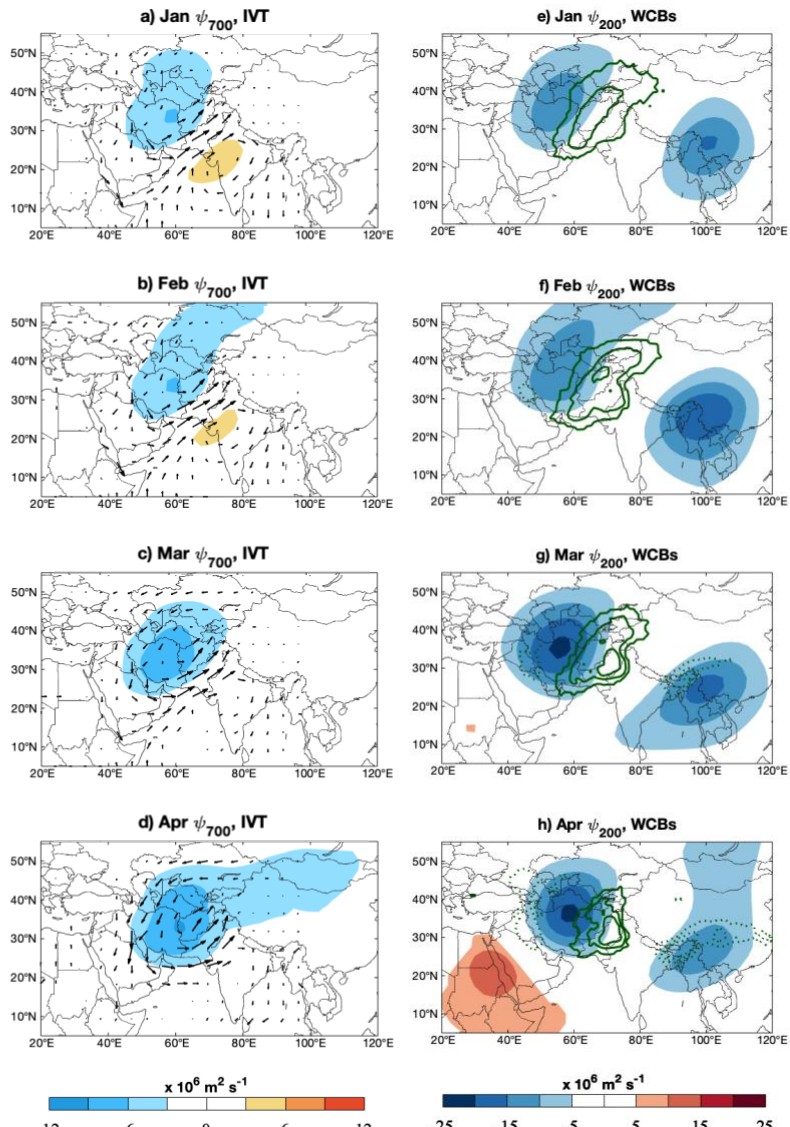

**Figure 7: Composite difference between 'heavy' precipitation days – meaning > 4 mm accumulation days - subtracted from 'light' precipitation days, when precipitation < 0.04 mm. For the months January – April, panels a) – d) show 700-hPa streamfunction ($\psi_{700}$) in the color shading and IVT in the vectors, and panels e) – h) show 200-hPa streamfunction ($\psi_{200}$) in the color shading and WCB frequency in the green contours, contoured at for values (-0.25, -0.15, -0.05, 0.05, 0.15, 0.25), with positive frequencies (solid) and negative frequencies (dotted). WCB units are events/6-hourly time step. Only differences that are statistically significantly different from random chance at the 95th confidence level, determined using bootstrapping with replacement, are shown. The number of light precipitation days in each composite is: 278 (Jan), 198 (Feb), 121 (Mar), 105 (Apr), and the number of heavy precipitation days is: 122 (Jan), 164 (Feb), 203 (Mar), 106 (Apr).**

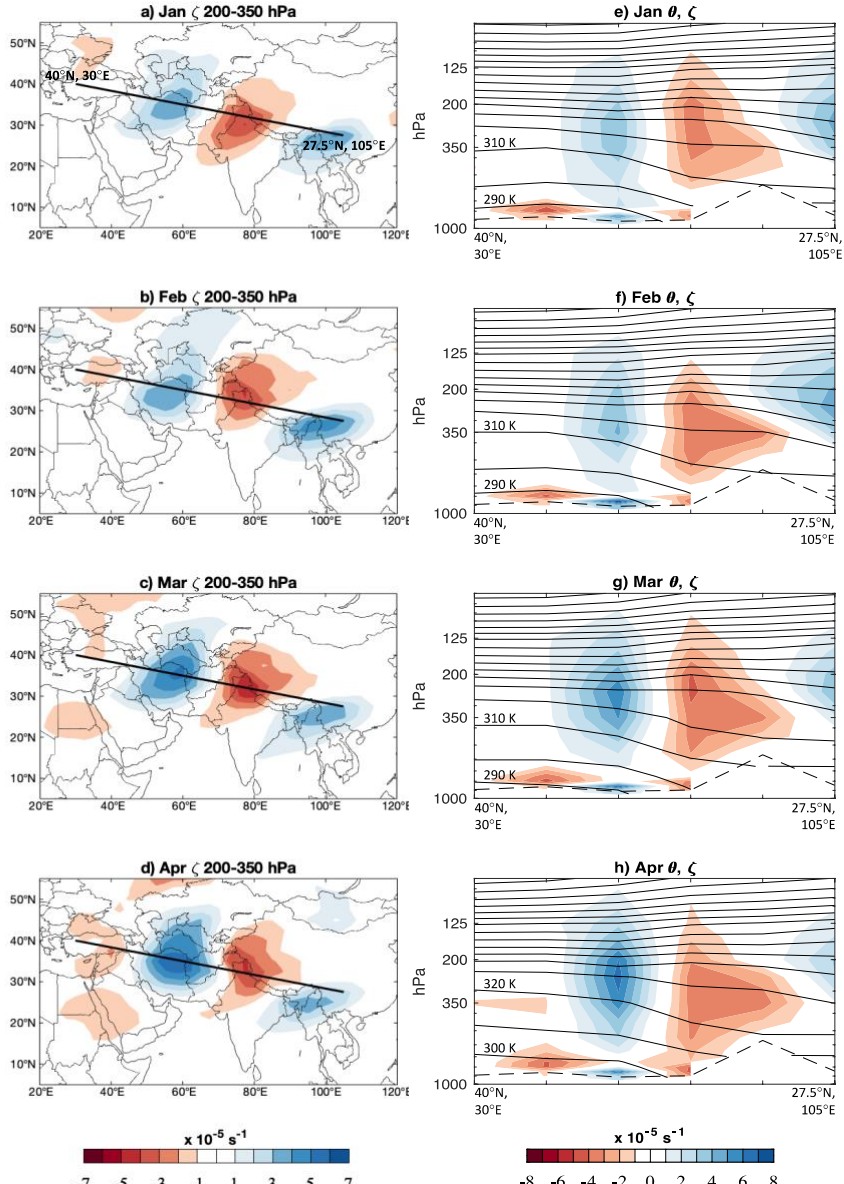

Figure 8: Panels a) – d) the color shading shows the 200-350 hPa mean relative vorticity, $\zeta$, difference, wet – dry days, for a) January, b) February, c) March, d) April, and the thick black lines show the location of the cross sections in panels e) – h). The color shading in panels e) – h) show the mean $\zeta$ difference, wet – dry days, and the black contours show the composite potential temperature, $\theta$, on wet days. Units of $\zeta$ are $10^{-5}$ s$^{-1}$, and units of $\theta$ are K, contoured every 10 K. The think dashed black line is the location of topography along the cross section.

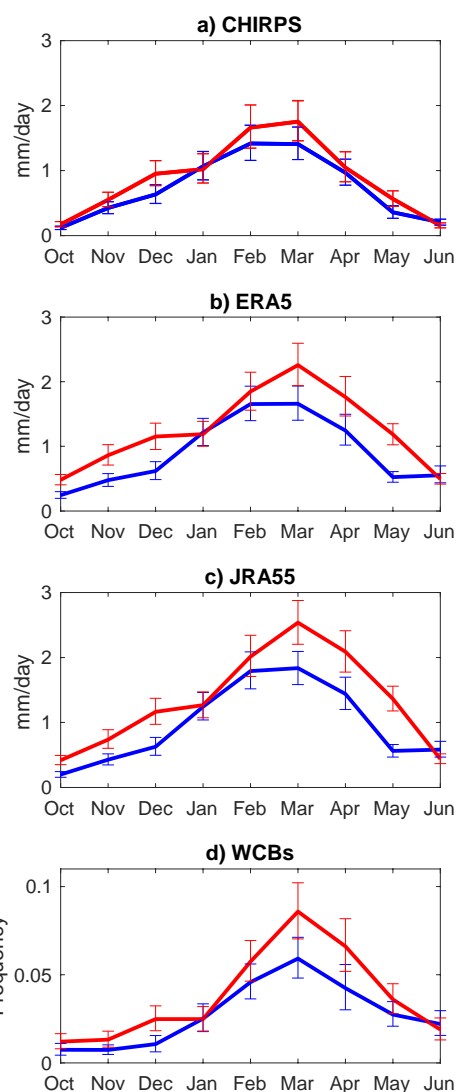


**Figure 9: Mean precipitation during El Niño (EN; red) and La Niña (LN; blue) months, determined using the ONI index, in a) CHIRPS, b) ERA5, c) JRA55 precipitation. Panel d) shows the mean WCB frequency (units events/6-hourly time step) during EN (red) and LN (blue) months. The error bars denote confidence at the 95% confidence interval, determined using bootstrapping with** 680 **replacement.**

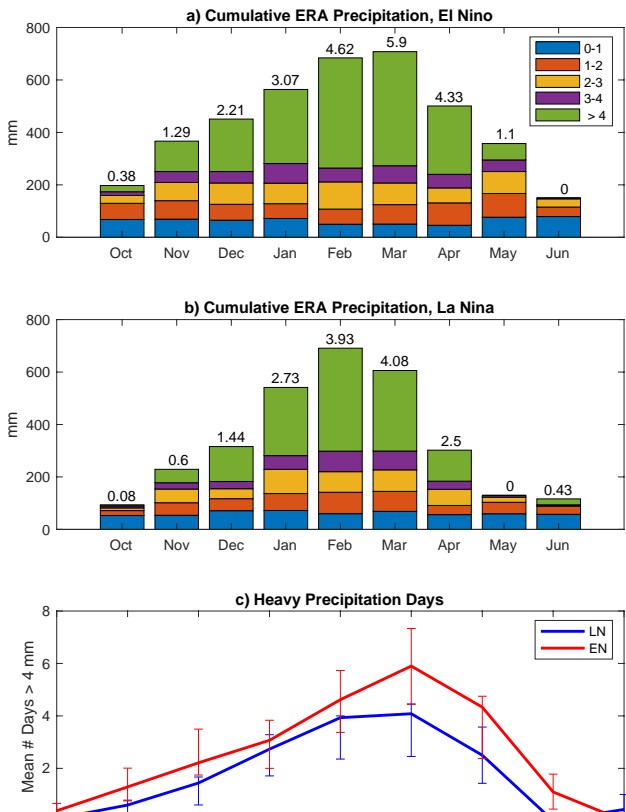

**Figure 10: Panels a) and b) show cumulative ERA5 precipitation binned by daily rate during El Nino and La Nina months (Table S1) that occurred from 1981-2017, respectively. The average number of days per month with > 4 mm totals are shown at the top of each bar, and these values are repeated in panel c). The error bars in panel c) denote confidence at the 95% confidence interval, determined using bootstrapping with replacement.**


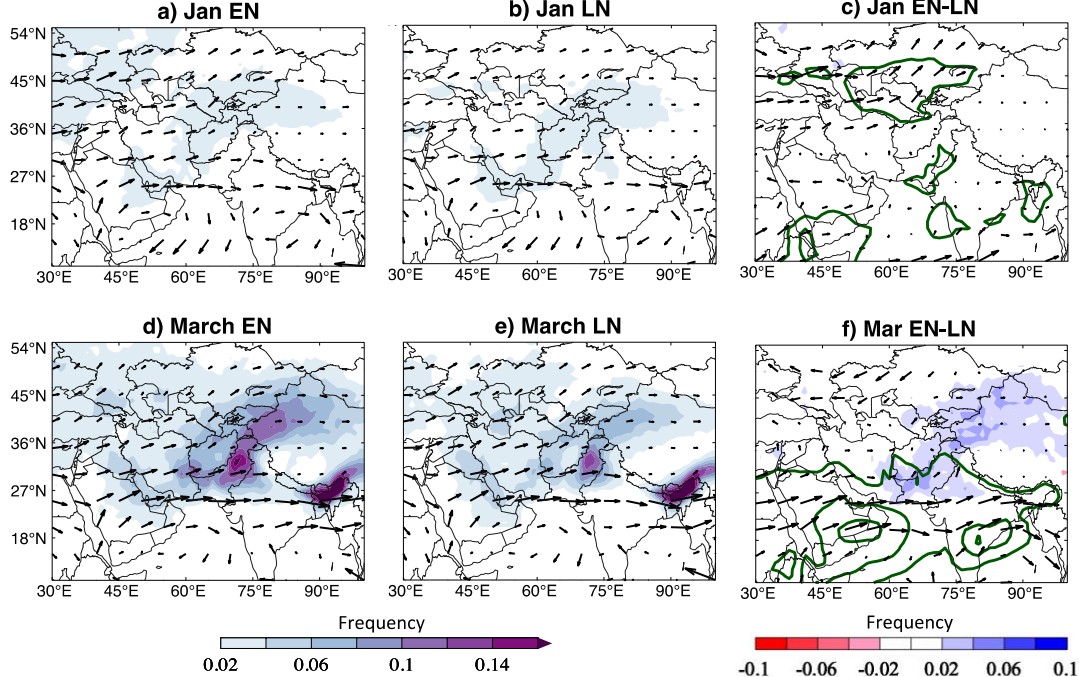


**Figure 11: The color shading in a) b) d) and e) shows the mean WCB frequency (units events/6-hourly time step), and the black arrows show mean IVT, in January and March during a) El Niño and b) La Niña months. Panel c) and f) shows the January and March difference, El Niño – La Niña, respectively, while the arrows show the difference in mean IVT. The green contours in panels c) and f) show the difference in precipitable water, starting at 1 kg m$^{-2}$ every 2 kg m$^{-2}$.  In panels c) and f), only differences that are**

**statistically different at the 95% confidence interval are shown.**

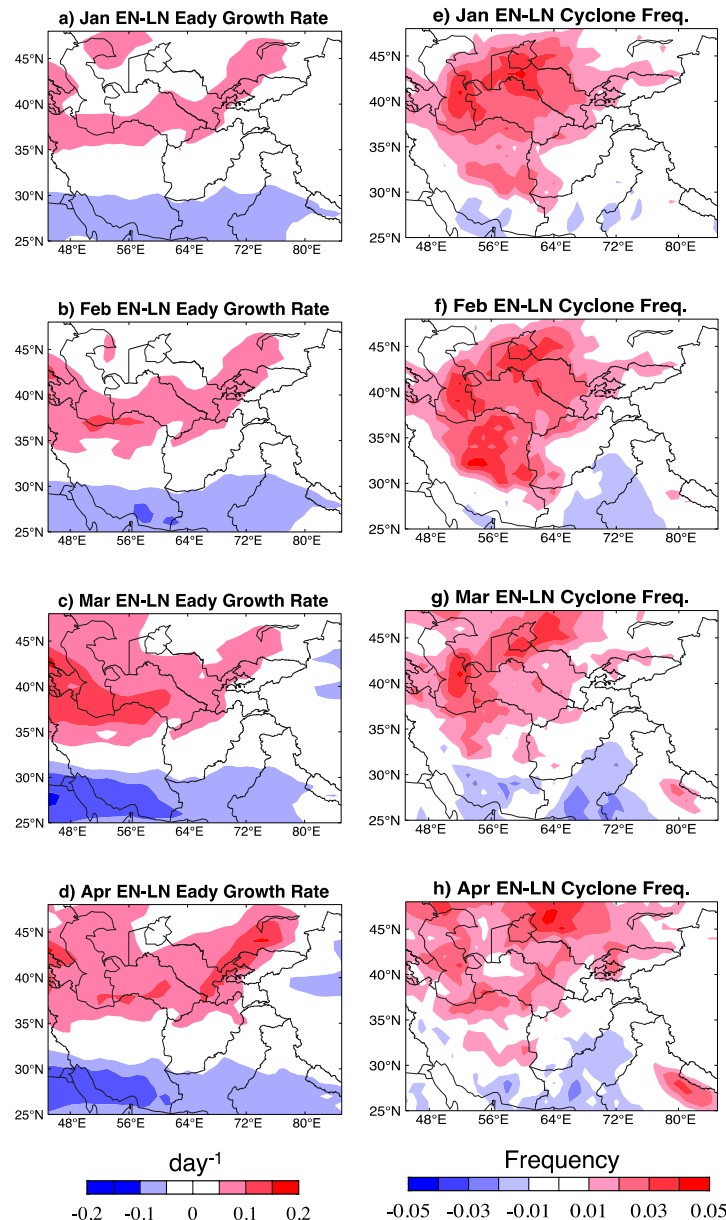

**Figure 12:** Panels a) – d) show the difference, El Niño (EN) – La Niña (LN), in the monthly mean Eady growth rate for the years 1979-2014, and panels e) – h) show the difference in cyclone frequency. Units of Eady growth rate are day⁻¹ and units of cyclone frequency are in units of events per 6-hourly timestep.