# Peer review of "The monthly evolution of precipitation and warm conveyor belts during the central southwest Asia wet season"

_EGUsphere, 2023_

## Author Comment (AC1)

Response to Reviewers
**Reviewer Comment 1**

Review of "The evolution of precipitation and warm conveyor belts during the central southwest Asia wet season" by Melissa L. Breeden, Andrew Hoell, John R. Albers, and Kimberly Slinski

**General comments:**

The present study by Breeden et al. investigates the seasonal evolution of precipitation over southwest Asia and the link of precipitation to warm conveyor belts and ENSO. The study first extensively compares different precipitation data sets and finds good agreement in terms of the timing of the peak precipitation. However, the data sets disagree concerning the precipitation amounts. The study continues with an analysis of the seasonal evolution and the comparison of days with little and heavy precipitation. Overall, the methods are sound, the text is well written, and the figures are clear. However, many of the conclusions remain rather vague and it is not upfront clear what the main novelty of the study is. Suggestions for further investigations that may help to strengthen the conclusions and to make this study distinct from other previous studies are given below.

Thank you for your careful attention to our manuscript. We have modified the text to better highlight the novelty in this manuscript, which is several-fold:

1) We include a month-by-month comparison of daily precipitation statistics between recent datasets that are used to study CSWA precipitation, including the comparison with the recently-available IMERG precipitation product.

2) We show the month-by-month evolution of WCB frequencies over CSWA during the wet season, which to the authors' knowledge has only been shown for the seasons December – February and March – May together, which are groupings that are not aligned with the precipitation maximum from February – April (this is not a critique of past literature, but a month-by-month analysis seemed a beneficial, if not incremental, addition to the published literature on this region).

3) We show the composite daily circulation differences between high and low precipitation days for the individual months from January – April, which has not been shown in the present literature, particularly, there has been an exclusion of consideration of April in several studies that have focused on the cold season from November – March. Figure 7 shows that the upper and low-level circulation patterns associated with heavy precipitation differs within the January – April period, so, we do feel we have shown how characteristics of the systems WCBs are associated with evolve during the wet season.

4) To the author's knowledge the month-by-month differences in CSWA WCB frequencies during La Niña and El Niño conditions have not been presented in the present literature. We also show how the number of precipitation days with > 4 mm accumulation drive the mean precipitation differences during El Niño versus La Niña conditions, information that prior studies using monthly data have not shown.

5) With the addition – at the reviewer's suggestion – of further exploring the reason for the observed ENSO/WCB differences, we find that the frequency of surface cyclones increases during El Niño

conditions, as does the Eady growth rate, consistent with the increase in WCBs and enhanced precipitation. However, while WCBs and precipitation do not change as a function of ENSO phase in January, cyclone frequencies do change, suggesting that cyclones and frequently used 'westerly disturbances' cannot always be used to infer precipitation changes in this region.

**Major specific comments:**

1) I fully understand the authors' motivation to investigate the link of WCBs to precipitation since these are one of the most important rain producing system in midlatitudes. That being said it seems that other weather systems may be of similar or even larger importance in CSWA. For example, the climatological WCB frequency in Fig. 6 is mostly less than 8% and even during the 'wet' days the difference is only on the order of 25%. This makes the reader wonder which other rain producing systems are responsible for precipitation in the region (especially during wet days). **At least a critical discussion of the potential role of other weather systems should be included in the manuscript.**

We understand this point, but noting that the warm conveyor belts identified are found to develop under different mean states and with different large-scale circulation patterns (Figure 7), we find the term to be a rather general representation of intense upward vertical motion that is not necessarily tied to a particular mechanism, as long as a surface cyclone is nearby. Pfahl et al. 2014 showed that the fraction of precipitation associated with WCBs *annually* is >50% over CSWA (their Figure 7b), justifying our focus on WCBs' role in CSWA precipitation. The % contribution is even greater for extreme precipitation events, which contribute heavily to the > 4 mm / day events we highlight in Fig. 3. We do show that the large-scale circulation pattern associated with WCBs and precipitation evolves during the wet season (Figure 7), confirming there are differences in how WCBs are formed at different times of the year. We are not sure exactly what 'other weather systems' means, and the reviewer unfortunately did not include any examples of weather systems that our method excludes. That said, we have modified the text to acknowledge that not 100% of precipitation is associated with WCBs in this region.

Regarding the frequency values of WCBs that the reviewer suggests are low, we consider an increase in WCB frequency to values of 0.25 (a tripling of the climatological frequency) to be significant – including statistically significant – indicating an average of 25% of trajectories at that location, that originate in the lower troposphere and have undergone 600-hPa of ascent within two days near a surface cyclone. It is easy to imagine that many trajectories during heavy precipitation days are ascending strongly might be *close to*, but don't *quite* meet, such specific requirements. One could imagine a progression in which 1) an amplifying cyclone would start to produce precipitation from ascent that is not intense enough to meet the WCB criteria, 2) the WCB criteria is met most frequently during peak ascent and precipitation, and 3) as the cyclone weakens, the ascent no longer meets the WCB criteria but may still produce precipitation. **We therefore do not expect WCBs to capture every single trajectory that involves condensation and precipitation, but rather to highlight the most intense ascent with the understanding that weaker or gentler ascent associated with the same cyclone over may still contribute to precipitation.** Given these considerations and the results of Pfahl et al. 2014, we feel the link between precipitation and WCBs is a salient connection. Admittedly, compared to other regions of the globe such as the north Pacific and north Atlantic storm tracks, WCBs are less frequent in

CSWA, but that fact is consistent with the semi-arid climate and the intermittent nature of precipitation in this region.

If we had omitted a crucial 'other weather system', we doubt the seasonality of WCBs and precipitation would align so clearly (indeed several of the metrics we considered first, such as moisture flux convergence and the upper-tropospheric storm track, did not display quite the same seasonality as precipitation).

Modified text (section 3.2): This is consistent with the results of Pfahl et al. 2014 who found annual CSWA precipitation was heavily related to WCBs (with >50% annual precipitation associated with WCBs over CSWA) and shows that this correspondence is greatest in late winter and early spring. We acknowledge that a portion of precipitation not associated with WCBs is observed in this region, though given the several requirements to identify WCBs, it is easy to imagine that some ascending trajectories associated with extratropical cyclones come close to, but do not quite meet, the WCB criteria. WCBs may therefore be considered as a reflection of the time of maximum ascent associated with extratropical cyclones, with the understanding that not every trajectory experiencing ascent and condensation may not be captured by this one definition.

2) The authors conclude that the modulation of WCB frequency by ENSO clarifies "the link between low-frequency circulation changes that ENSO produces, and the transient, short-lived nature of precipitation in this region". Though I generally agree with this interpretation it remains unclear whether it is the WCB frequency alone that leads to a reduction of precipitation or whether the WCBs that occur are associated with increased/reduced precipitation. To answer this question the authors may want to adapt an established approach (e.g., Catto et al. 2012, Hauser et al. 2020) that decomposes rainfall anomalies into changes in intensity and frequency.

Thank you for this suggestion, we agree that the question of whether individual WCBs and precipitation events are more intense during El Niño compared to La Niña conditions, or if WCBs are occurring more frequently at similar intensity, is an interesting one. To examine this, we have compared the median and $95^{th}$ percentile values of WCB frequencies between EN and LN conditions during the months February-March-April (when WCB differences are greatest), revealing that the median WCB frequency does shift significantly towards more WCBs during El Niño than La Niña (Figure R1, also Figure S3 in modified manuscript). Conversely, the $95^{th}$ percentile values of WCB frequency do not shift significantly, implying that it is the frequency of WCBs of similar intensity, rather than the formation of *more* intense WCBs, that are associated with EN conditions and enhanced precipitation. We have modified the text to include this information and added the figure below to the supplement, and included the interesting suggestion to use the decomposition of Catto et al 2012 and Hauser et al. 2020 in the discussion as a potential avenue of future work.

[Figure]

Figure R1 (now Fig. S3): The left panel shows the bootstrapped median WCB frequency values averaged over Afghanistan (box in Fig.1 in manuscript) during El Niño (red) and La Niña (blue) months, and the right panel shows the 95th percentile WCB frequency values.

Modified text in section 3.3: Finally, to explore whether WCBs are more intense or more frequent during EN months, we compare the median and 95th percentile values of WCB frequency for EN/LN conditions for the months February-March-April (Fig. S3). The median WCB frequencies are statistically significantly different, while the 95th percentile values are indistinguishable, implying that the strongest WCBs are not changing in frequency as much as WCBs of average intensity.

Modified text in section 4: Future work could decompose precipitation ENSO-related precipitation changes into intensity and frequency contributions following Catto et al. 2012 and Hauser et al. 2020, including the lack of a response in January, which may be related to competing effects of intensity and frequency changes.

3) Though the statistical link between ENSO and WCBs is clear, it remains unclear what the physical connection may be. Is the number of WCBs increasing during El Nino due to planetary Rossby waves excited by the tropical convection? Is the number enhanced due to higher SST that increase moisture in WCB inflow regions? Some explanation and analysis which **establishes the dynamical link between ENSO and WCB activity in CSWA would definitely help to strengthen the conclusions**. This is particularly necessary as other studies have investigated dynamical links between ENSO and CSWA rainfall (see introduction) and it is not clear what additional insight the present study provides.

We agree that further understanding the physical mechanism(s) associated with changes in WCB frequency during different ENSO phases is important to address, though we struggle to agree with the statement that additional insight is not present already – please see prior our response outlining the novel, though sometimes straightforward, results that are presented in this manuscript. One example is our finding that a small number (<20% of days) of intense precipitation days largely drives the mean precipitation changes associated with EN / LN conditions is a novel result that has not been shown by prior monthly-mean or annual-mean analyses.

That said, we agree that the section could be expanded, and have calculated the composite difference in the monthly mean Eady growth rate (Eqn. S1) and cyclone frequencies between La Niña and El

Niño months for the years readily available online (1979-2014) from the same ETH Zürich ERA-Interim Lagrangian Climatologies database used for the WCBs (http://www.eraiclim.ethz.ch/). Both the Eady growth rate and cyclone frequency increase substantially just east and north of Afghanistan during EN compared to LN conditions (Figure R2), which is broadly consistent with WCBs on their eastern flank being located over Afghanistan, increasing precipitation. The changes in Eady growth rate and cyclone frequency are hypothesized to be related to changes in the mid-level vertical shear that is one of the terms in Eqn S2. The increase in the growth rate and cyclone frequencies in this area provides further insight regarding how El Niño increases the occurrence of WCBs and precipitation over Afghanistan, although we note that there are cyclone increases in January when no WCB/precipitation changes are observed, implying cyclone frequency changes are not always reflective of WCB or precipitation changes (this was also noted by Joos et al. 2023). We have added a discussion of these results to Section 3.3 and thank the reviewer for the suggestion.

[Figure]

Figure R2 (now Figure 11): Panels a) – d) show the difference, El Niño – La Niña, in the monthly mean Eady growth rate for the years 1979-2014, and panels e) – h) show the difference in cyclone frequency. Units of Eady growth rate are day$^{-1}$ and units of cyclone frequency are in units of events per 6-hourly timestep.

Joos, H., Spenger, M., Binder, H., Beyerle, U., and H. Wernli: Warm conveyor belts in present-day and future climate simulations – Part 1: Climatology and Impacts, Weather Clim. Dynam. 4, 133-155, 2023.

**Minor specific comments:**

Title: To me, it is not upfront clear that "the evolution" means "the seasonal evolution". My suggestion is therefore to specify that the seasonal evolution is meant. Also, I am wondering whether it would be beneficial to specify that the study also considers quite extensively the variability of precipitation. Thank you for this suggestion, we agree and have included 'monthly' to the title.

l. 14: I find it confusing that WCBs are chosen to represent "local vertical motion forcing". In my view, WCBs are rather a result of local vertical motion forcing than a representative of forcing. Perhaps just leave out the word "forcing"? Thank you for raising this point, we have removed 'forcing'.

l. 19: The sentence "Precipitation intensity, duration, and the associated circulation patterns evolve...." remains quite unspecific. Be specific on how exactly they evolve/change as winter progresses. Clear statements will make it easier to identify the main conclusions and likely increase the visibility of the study. We added a description of how the patterns evolve.

l. 19: It would be helpful to indicate that it is the area-mean daily accumulation. We have included the descriptor 'area-averaged'.

l. 21: What is meant with "heavy precipitation days"? Is this referring to days when the accumulation exceeds 4 mm? Please specify. Thank you, we have specified > 4 mm/day.

l. 23: The reader may wonder what is meant with "neither precipitation nor WCB change". Does this refer to intensity, duration or frequency changes? Please clarify. We meant mean precipitation which has now been specified.

l. 45: Rather use em-dashes than en-dashes. Done.

l. 73: Is past-tense used on purpose here? In my view, present tense could be used in this and the following sentences as well. Thank you, we changed the text to present tense.

l. 87 and elsewhere: Use en-dashes to indicate ranges between two values. Thank you, we have made these changes.

l. 87: Are the precipitation data sets used at their native grid spacing or is any kind of remapping performed? For this study we kept the datasets at the grid spacing provided to us. CHIRPS and IMERG are available at higher horizontal resolution, but are also provided at the resolutions used. No remapping was performed by the authors.

l. 89: Can you specify how many rain gauges are actually located in Afghanistan? "Relatively few" is rather vague. The number of gauges varies in time between a minimum of zero to under 60 : https://data.chc.ucsb.edu/products/CHIRPS-2.0/diagnostics/stations-perMonthbyCountry/pngs/Afghanistan.066.station.count.CHIRPS-v2.0.png. We have added the range to the text.

l. 92: Is ERA5 (and JRA55) really incorporating satellite-derived precipitation estimates? To my understanding, ERA5 precipitation is based on short-range forecasts of ECMWF's IFS system. Thus, the total precipitation in reanalyses is the sum of large-scale precipitation generated by the cloud scheme and convective precipitation generated by the convection scheme. To my knowledge precipitation estimates are only assimilated over the United States from 2009 onward. Thank you for raising this point. We meant that in the creation of the ERA5 and JRA55 datasets, the data assimilation process includes satellite observations, but mistakenly included the word 'precipitation' as you noted. We have fixed the text.

l. 99: Can you provide some specific information concerning the number of stations in the country?: The number of gauges varies in time between a minimum of zero to under 60 : https://data.chc.ucsb.edu/products/CHIRPS-2.0/diagnostics/stations-perMonth-byCountry/pngs/Afghanistan.066.station.count.CHIRPS-v2.0.png. We have added the range to the text.

l. 103: What is the motivation for using a coarser resolution for streamfunction than for IVT? Streamfunction is by nature a smoothed field given the way it is attained by integrating the velocity field. The horizontal resolution used was based on the availability of local data that was used to avoid redundant datasets. We do not think the resolution impacts our results in any way.

l. 115: It would be very helpful to provide this information right at the beginning of section 2.1. Otherwise, the reader may wonder what the exact time period analysed in this study is. Thank you, we have moved this statement to earlier in the text.

l. 119: Does "probability curve" mean "probability distribution"/"probability density function"? Yes, we have changed the wording from 'curve' to 'distribution'.

l. 135: Can you elaborate on why a coarser resolution would lead to higher means and standard deviation? Thank you for the question. As noted by the word 'potentially', this was a supposition based on a comment from a colleague, but we have opted to remove the text to avoid confusion as this is not a crucial point.

l. 143/Fig.3: Are the values accumulated over the entire period 1981-2020? If so would it not be easier to understand if you showed mean precipitation per month? The focus of this figure is to compare the relative values between each threshold, and the relative increases and decreases for each month, information which will not change if we divide the total accumulations by 40 to get the mean. Since the median and percentile values per month are shown in Figure 2, we prefer to keep the accumulation to offer a sense of the total amount of precipitation that occurs in this region, which a mean does not provide. This perspective is used in section 3.3 to highlight that more precipitation accumulates during fewer EN months compared to LN months.

l. 180: I had difficulties to find the corresponding figure in Pfahl et al. 2014 showing the correspondence of precipitation and WCBs during spring. Just out of curiosity could you indicate the corresponding figure? Figure 7b in Pfahl et al. 2014 show the annual contribution of WCBs to

precipitation. *Our* results indicate that this relationship peaks in late winter early spring. We have clarified the text.

l. 180: "heavily related" is a rather qualitative statement. Could you indicate what percentage of precipitation is roughly related to WCBs? My interpretation of Fig. 7b in Pfahl et al. 2014 is that roughly 50% of precipitation in Afghanistan is linked to WCBs. Please see our prior response regarding linkages between precipitation and WCB, and the intent and limitations behind using WCBs.

l. 247: Can you comment on how El Nino is linked to strengthened synoptic activity and moisture availability? I assume that global teleconnection patterns play an important role. Thank you, you are correct. We have now shown the differences in surface cyclone frequency and Eady growth rate (new Figure 11) which have addressed the question of synoptic activity, while Figure 10c and 10f show the precipitable water differences between EN and LN conditions, confirming that large precipitable water increases are observed during EN compared to LN months, most notably in March. We mention that the enhanced moisture near the inflow WCB region south of Afghanistan is likely indicating an increase in the moisture of WCB trajectories, increasing diabatic heating and precipitation. The explanation of the global teleconnection related to ENSO is already described in the introduction (see response below for the text as well).

l. 257: Please double-check the usage of past and present tense. Done.

l. 270: Can you comment on the moisture availability in this region which potentially effects the ascent of WCBs? Thank you for this suggestion, the increase in precipitable water during EN versus LN month, observed in February-March-April but not January, is certainly consistent with an increase in the low-level moisture availability, increasing the diabatic heating within the ascending airstreams of developing cyclones. By generating potential vorticity anomalies conducive to strengthening the cyclone (e.g., Madonna et al. 2014), ascent could be strengthened, increasing the amount of trajectories meeting the WCB criteria and precipitation. The fact that January, the month without WCB and precipitation changes, does not involve an increase in precipitable water near the inflow region, while other months share this characteristic, is particularly compelling. We have added this discussion to the text.

l. 278: As before it would be worthwhile to comment on how ENSO is dynamically linked to rainfall in CSWA. This text is copied from the introduction: "It has been well-established that SST variations and the associated tropical convective heating response can modulate CSWA precipitation through the generation of an anomalous Rossby wave source and large-scale vertical motions (Barlow et al. 2002; Hoell et al 2012; Hoell et al. 2013; Hoell et al. 2015b; Hoell et al 2018a; Hoell et al. 2018b). In particular, the El Niño-Southern Oscillation (ENSO) is a strong modulator of southwest Asian climate, with La Niña associated with reduced moisture fluxes (Mariotti et al. 2007) and an anomalous upper-level anticyclone and downward vertical motion (Nazemosadet and Ghasemi 2004; Hoell et al. 2014a; Hoell et al. 2014b; Hoell et al. 2015a; Breeden et al. 2022), both conducive to reduced precipitation that can lead to drought (Barlow et al. 2002). Conversely, anomalous rising motion and enhanced precipitation are observed during El Niño conditions (Hoell et al. 2017)."

l. 282: On a local scale I agree with this interpretation. But what circulation change modulates the occurrence frequency of WCBs in first place? Please see our prior responses and additional analysis

about the enhanced cyclone frequency, Eady growth rate and moisture content of the ascending airstreams during El Niño versus La Niña conditions.

References

Barlow, M., Cullen, H., and Lyon, B.: Drought in central and southwest Asia: La Niña, the warm pool, and Indian Ocean precipitation, J. Climate, 15, 697–700, https://doi.org/10.1175/1520-0442(2002)015<0697:DICASA>2.0.CO;2, 2002.

Pfahl, S., Madonna, E., Boettcher, M., Joos, H., and Wernli, H.: Warm Conveyor Belts in the ERA-Interim Dataset (1979–2010). Part II: Moisture Origin and Relevance for Precipitation, *Journal of Climate*, 27(1), 27-40, 2014.

**References:**

Catto, J. L., Jakob, C., and Nicholls, N. (2012), The influence of changes in synoptic regimes on north Australian wet season rainfall trends, J. Geophys. Res., 117, D10102, doi:10.1029/2012JD017472.

Hauser, S, Grams, CM, Reeder, MJ, McGregor, S, Fink, AH, Quinting, JF. A weather system perspective on winter–spring rainfall variability in southeastern Australia during El Niño. QJR Meteorol Soc. 2020; 146: 2614– 2633. https://doi.org/10.1002/qj.3808

**Reviewer Comment 2**
**The evolution of precipitation and warm conveyor belts during the central southwest Asia wet season**

This manuscript contains a study that focuses on wet-season precipitation over Afghanistan on a daily scale, investigating the association of heavy precipitation days with the presence of warm conveyor belts (WCBs) and the link with climate modes such as ENSO. The manuscript is well-written and the analysis is clear and logical. However, there are a few primary questions I would like to be addressed before it can be accepted for publication. Those questions are listed below, followed by minor line-by-line comments.

**Primary Comments:**

- My understanding is that the key novelty of this manuscript, that sets it apart from the previous papers on the topic, is the focus on a daily time scale, rather than monthly or seasonal, for precipitation. If this is the case, then a more detailed analysis needs to be performed on what the causes of precipitation variability at such a time scale are.

Thank you for your attention to our manuscript. We have modified the text to better highlight the novelty in this manuscript, which has several components:

1) We include a month-by-month comparison of daily precipitation statistics between recent datasets that are used to study CSWA precipitation, including the comparison with the recently-available IMERG precipitation product.

2) We show the month-by-month evolution of WCB frequencies over CSWA during the wet season, which to the authors' knowledge has only been shown for the seasons December – February and March – May together, which are groupings that are not aligned with the precipitation maximum from February – April (this is not a critique of past literature, but a month-by-month analysis seemed a beneficial, if not incremental, addition to the published literature on this region).

3) We show the composite daily circulation differences between high and low precipitation days for the individual months from January – April, which has not been shown in the present literature, particularly, there has been an exclusion of consideration of April in several studies that have focused on the cold season from November – March. Figure 7 shows that the upper and low-level circulation patterns associated with heavy precipitation differs within the January – April period, so, we do feel we have shown how characteristics of the systems WCBs are associated with evolve during the wet season.

4) To the author's knowledge the month-by-month differences in CSWA WCB frequencies during La Niña and El Niño conditions have not been presented in the present literature. We also show how the number of precipitation days with > 4 mm accumulation drive the mean precipitation differences during El Niño versus La Niña conditions, information that prior studies using monthly data have not shown.

5) With the addition of further exploring the reason for the observed ENSO/WCB differences, we find that the frequency of surface cyclones increases during El Niño conditions, as does the Eady growth rate, consistent with the increase in WCBs and enhanced precipitation. However, while WCBs and precipitation do not change as a function of ENSO phase in January, cyclone frequencies do change, suggesting that cyclones and frequently used 'westerly disturbances' cannot always be used to infer precipitation changes in this region.

WCBs are airstreams that are part of weather systems such as extratropical cyclones. Basing the analysis on WCBs without looking at what systems they are associated with makes the analysis incomplete.

It is argued in the conclusions that the analysis of cyclones' lifecycle and their changes throughout the wet season could be a future avenue of research, but I would rather suggest that this activity (even in an exploratory way) is used to complement the present manuscript. "Westerly disturbances" are mentioned at several points in the text. Are they the same western disturbances as in Hunt et al., 2017 (https://doi.org/10.1002/qj.3200) and following papers by the same first-author (and I should specify I am not him or any of his co-authors)? How does their structure and evolution changes throughout the course of the season?

Figure 7 addresses your question regarding how the structure of the large-scale pattern and WCBs changes within the wet season, with a transition from positively-tilted troughs January and February to, in April, a more neutral tilt, a shift in the WCB maximum to the east, and associated streamfunction anomalies over North Africa that are not observed during the heart of winter. The positive tilt observed in January and February is consistent with the mean structure of the westerly disturbances studies by Hunt et al. 2017, a link we have added to the text. In contrast to our study, which breaks down the composite circulation into month-by-month differences, Hunt et al. 2017 studied the structure of disturbances that occurred year-round, obscuring the month-by-month circulation differences that we have shown are related to precipitation. Hunt et al. 2017 also

considered disturbances that tracked farther to the south and east compared to our focus region (see their Fig. 1b for the domain of interest), which include an area that displays differences in precipitation seasonality than Afghanistan (see their Fig. 1a). Still, there are indeed similarities between the characteristics of the disturbances they studied and the features examined here, which we have now mentioned in the text, while we emphasize that the frequency of cyclones as opposed to WCBs/precipitation is not one-to-one, as shown in the newly-added Fig. 11 and supported by prior studies, indicating the changes of cyclones during El Niño and La Niña conditions are more consistent from month-to-month than the location of WCBs and precipitation. This is important because the WCB changes, not the cyclone changes, best align with precipitation seasonality and changes as a function of ENSO.

Without answering these questions, analysing WCBs alone leaves many questions open on what mechanisms could be responsible for the changes in precipitation shape between months. For example, looking at Figure 5, the frequency of WCBs over Afghanistan in Feb and April is very similar. There must then be other factors behind the different behaviour in precipitation.

Thank you, yes, we have highlighted the differences in the circulation patterns associated with WCBs and precipitation with the composites in Figure 7 (see previous response). Recall that Section 3.2 focuses on the heavy (> 4 mm / day) events, which do not display notable differences in their frequency or duration, comparing January and April (Fig. S1, Fig. 4, February's distribution is similar to January). The marked changes in the persistence of precipitation we highlight are evident at much lower accumulation values than 4 mm/day, which is interesting (hence our mentioning it) but not the focus of the subsequent analysis in Section 3.2. In terms of the frequency of *heavy* precipitation events, CHIRPS, JRA55 and ERA5 all show agreement regarding the minimal differences between February and April. We have added a clarifying statement to the text.

- As said above, it is argued in the manuscript that heavy precipitation is associated with WCBs, which are linked to systems such as extratropical cyclones and to the pattern of the upper-level sub-tropical westerly jet. Therefore, without discounting the importance of ENSO, it would make sense investigating also the link with climate modes more directly related to the Eurasian circulation at these latitudes (NAO, CGT, …).

We focus on ENSO for several reasons, including the robust relationship between CSWA precipitation and ENSO found in prior studies, because it is the most predictable climate oscillation, and because it is by nature, a low-frequency oscillation that consistently modifies the background state in the same manner for the duration of the month. There is a wealth of prior literature referencing the teleconnection between ENSO and CSWA precipitation (see introduction), more so than other teleconnections, which is why we focus on ENSO in particular. At the suggestion of the other reviewer, we have opted to include some additional analysis of the El Niño / La Niña differences (see new Figure 11). Still, we have added a mention of additional teleconnections that may be of interest for future work in Section 4.

Regarding other teleconnections such as the North Atlantic Oscillation, the link to precipitation in the broader region seems to be more tenuous (see discussion in Hunt et al. 2022) and strongest farther to the east of the domain of interest as shown by the figure from Hunt et al. 2022 for the years 1950-2015, copied below. We thus find the current literature further substantiates our focus on ENSO but nonetheless, we have mentioned the NAO briefly in the introduction.

[Figure]

**Fig. 4** Effect of the NAO on Dec–Mar precipitation and vertically integrated moisture flux over South Asia. **a** Dec–Mar climatology, **b** composite difference between NAO+ and NAO− months and **c** the ratio of the difference and the climatology. The black rectangle in **a** marks the same region as in Fig. 3. Stippling in **c** indicates where a Welch's *t* test indicates where the distributions of NAO+ and NAO− precipitation are significantly different at the 95% confidence level

Reference: Hunt, K. M. R. and S. N. Zaz: Linking the North Atlantic Oscillation to winter precipitation over the Western Himalaya through disturbances of the subtropical jet, Clim. Dyn., 60:2389-2403, https://doi.org/10.1007/s00382-022-06450-7, 2022.

\- Given the location of the area under study, with several mountainous areas in it and at its edges (including the very high Hindukush – Karakoram range), it would appropriate to consider the impact of orographic effects on precipitation.

We agree that orographic enhancement in the presence of upward vertical motion forcing is an important factor that enhances precipitation in this region, and we mention this in the first paragraph of the introduction. However, given the distinct seasonality of precipitation, the presence of mountains is clearly not a sufficient condition to produce precipitation. The intent of the paper is to survey the seasonality of intense vertical motion associated with surface cyclones, here represented by WCBs, which has not been exhaustively explored over CSWA in the literature (see prior response regarding the new information in this manuscript).

\- Wet-season precipitation in the area is often described as sporadic. However (lines 147-148) is having up to 6 days per month with precipitation over 4 mm really "a reflection of the intermittent and intense nature of precipitation" and of it being sporadic ? It does not seem a small number to me. Could you help putting this into context by comparing values against other regions in the world where precipitation is known to be less intense, intermittent, and sporadic and more regular and frequent? (You could start doing this in the Introduction by looking at relevant literature and then use that discussion as a reference while commenting the results).

Thank you, we meant sporadic to indicate intermittent, reflecting the role of short-lived and intense precipitation days on mean precipitation as demonstrated by Figure 3 and Figure 4. We have removed the word from the text to avoid confusion.

**Minor comments, line-by-line:**

Line 36: We are moving beyond the science here, but looking at the recent history I don't think "humanitarian food aid is provided to the country [Afghanistan]" solely "given the crucial but unreliable role that precipitation plays in the region". A few words mentioning general geopolitical context would help. Thank you, certainly, we did not mean to imply intermittent precipitation is the sole driver of humanitarian aid. We have modified the text accordingly.

Line 44: Is this statement referred to precipitation globally or in the CSWA region? Please specify. We have specified we meant CSWA regarding in situ precipitation observations.

Line 52: I am not disputing the expertise of the authors of the present paper, but it would be appropriate to add here more studies that they are not authors of. At the same time, I would suggest considering more teleconnections (see related general comment).

The seasonality of precipitation changes dramatically between Afghanistan and the western Himalayan region covering Pakistan and northern India due to the influence of the summer monsoon, as previously mentioned, making it complicated to draw linkages between studies that focus on these physically nearby but distinct hydroclimatic areas. We have made an effort to include as many relevant references to our particular area of study as we could find, including several Hunt et al. references at the reviewer's suggestion.

Given the authors' history conducting research for FEWS NET, which has had a long-standing interest in this area, it follows that many of the relevant references are related to the authors. Still, we have revisited our literature search for relevant papers, leading to several references being added to the text. We now feel that our literature review has been sufficiently thorough.

Please see our prior response regarding our decision to focus on ENSO, and recall that only one of three sections in the paper addresses teleconnections, because exploring teleconnections is not the sole objective of the present analysis, but was determined a useful exploration the seasonality of WCBs and precipitation were found to align. We have mentioned the NAO / Western Himalayan precipitation link in the introduction – noting that the literature has not agreed on the NAO influence to the broader region, and that a recent analysis has shown that the NAO influence is farther to the east than Afghanistan – and that future work could address additional teleconnections.

Line 74-81: I would limit this paragraph to a short summary of what the following sections contain, without mentioning the results here (such a discussion would better fit to the "Discussion and conclusions" section). Thank you for the suggestion, we have shortened this paragraph.

Line 104: Please define IVT (What's its mathematical expression? What levels are used? …) or point to a reference. Also, the word "'horizontal" does not seem to be in the right place here. If it indicates that moisture flux is calculated using horizontal wind, I would insert it between "integrated" and "moisture". Thank you for these questions – yes, IVT is a vertical integration of the horizontal moisture flux. We have moved the word 'horizontal' as suggested and included the expression for zonal and meridional IVT as Eqns 1 – 2, respectively.

Line 130 - Fig 1: Mountain ranges near Afghanistan (Hindukush – Karakoram to the E and Pamir, I think, to the N) are characterised by large precipitation amounts, as shown in the panels of this figure. What are the reasons for not including them in the domain under study? I understand your focus on semi-arid Afghanistan's climate and I am not asking you to widen the domain, but I think

that a short discussion on whether precipitation in those mountainous areas is then collected in river basins crossing Afghanistan or not should be included. Thank you for raising this point. Pakistan has a different precipitation seasonality than Afghanistan (bi-modal rather than uni-modal) due to Pakistan's closer proximity to the summer Indian monsoon, which only reaches far southeastern Afghanistan in some years, but not all (see Figure below from Funk et al. 2015). These differences are the motivation behind our domain selection. Nonetheless, the composites in Fig. 7 indicate that the events associated with area-mean heavy precipitation over Afghanistan are associated with WCBs and precipitation over a wider area than that selected for the area-averaging. We have included a statement explaining our choice of domain for these reasons to section 3.1.

[Figure]

Figure: Map showing the wettest three-month seasons based on CHPclim (a high-resolution precipitation climatology with reference Funk, C. et al. A global satellite assisted precipitation climatology. Earth Syst. Sci. Data Discuss 7, 1–13 (2015)).

Figure reference: Funk, C., Peterson, P., Landsfeld, M. Pedreros, D., Verdin, J., Shukla, S., Husak, G., Rowland, J., Harrison, L., Hoell, A., and J. Michaelsen: The climate hazards infrared precipitation with stations – a new environmental record for monitoring extremes, Sci Data, 2, 150066, https://doi.org/10.1038/sdata.2015.66, 2015.

Figure 2: CPC is not mentioned in the caption. Thank you for pointing this out, we have made the correction.

Figure 3: Please increase font size in legends, axes, and on top of bins (no more than 1-2 decimal digits are needed), as numbers are difficult to read. Thank you for the suggestion, we have made these changes.

Figure 3: I would say "average number of days per month" rather than "per year" in the caption (as it is already clear that this is over 1981-2020). We have made this change.

Figure 3: Please give more details on what "cumulative" means. Is this cumulated over all years and averaged over the area? Or cumulated over the area and average over all years? Or something else? We first take the area-average in the box in Figure 1 and then sum over all years of daily data. We have made this explicit in the Figure 3 caption.

Lines 154-155: What is the definition of an individual precipitation event? How long does the no-precipitation time has to be to separate two distinct events? Events are characterized by strings of consecutive days meeting the required threshold. One day of separation is required between events, which is justified given the low autocorrelation timescale ( < 0.5) of area-mean precipitation during the broader wet season from October – April. We have modified the text in Section 2.2 accordingly.

Line 169: Where do I see 12 mm accumulation in Fig 4? There does not seem to be a "total event accumulation" contour in the figure The third black dashed line is the 12.2 mm total accumulation line. The red probability contours in Fig. 4b and Fig. 4c indicate the presence of events that exceed that amount of accumulation, since they are to the right of the black dashed 12.2 mm contour.

Figure 7: in the caption, shouldn't it be "contoured at +/- 0.05, starting at 0.1"? Otherwise, I am not sure I understand what WCB frequency values are shown. Also, it is rather difficult to see the green dashed contours. As was written, we begin contours at values of +0.05 and -0.05, and contours increase and decrease from those intervals, respectively, at intervals of 0.1. In other words, we contour WCB frequency values of -0.25, -0.15, -0.05, 0.05, 0.15, 0.25. We do not, as you suggest, contour values -0.2, -0.15 -0.1, 0, 0.1, 0.15, 0.2. We have listed the contour values precisely to avoid confusion.

Line 224: Is it "lack of significant tropical Pacific SST anomalies associated with…"? Thank you, we have added the word 'anomalies'.

---

## Author Response (AR2)

Second Review for WCBs paper – Minor revisions
Dear Authors,

Many thanks for spending time towards addressing my comments. I am happy with most of the responses although unfortunately I am still not convinced by all of them.

In particular, a key area for improvement in the manuscript is still the description of the cyclones (presumably western/'westerly' disturbances) associated with WCBs. I don't think Fig7 goes far enough in showing their features. As I said in my previous review, if one of the novelties of this study resides in the month-by-month focus, then greater emphasis will need to be placed on the characterisation of short-timescale features affecting precipitation. WCBs are certainly part of the story, but without a link to the cyclones they're associated with the analysis does not seem complete to me.

I wouldn't want to this to be too much of a stumbling block for the acceptance of this otherwise nice work, so I would just ask you to add a couple of figures showing the horizontal and vertical structure of those cyclones, for example along the lines of https://doi.org/10.5194/wcd-2-1303-2021, figs 9-10. They used ERA5 data so you should have access to the same fields. You won't necessarily have track locations for the cyclones as they do, but you could just centre your plots (let's say) 500km west of WCB locations. You could just focus on one time (heavy prec days, possibly subtracting light prec days as in your fig7) instead of the whole lifecycle as they do, and compare cyclone structures between different months.

Without doing this (or equivalent analysis) it seems difficult to me that you can go beyond just inferring causes of WCB development (as you do at lines 214-217) and shed light on what drives their frequency/strength/shape, which I think would substantially improve the manuscript and, in my view, bring it towards acceptance.

Response: We greatly appreciate your continued attention to our manuscript and the clarification for what additional information would benefit the manuscript. Using the reference you provided as a guide, we have included a new figure including cross sections of the relative vorticity and potential temperature fields, in addition to modified text (copied below, now lines 227-250 and Figure 8). We believe that showing the horizontal and vertical structure of the vorticity differences and associated streamfunction effectively illustrates the equivalent barotropic structure of the circulation associated with heavy precipitation and highlights how the strength and depth of the upper- and lower-level vorticity differences change from month to month.

New text:
'Vertical cross sections of the relative vorticity difference between wet and dry days, taken through the main dipole comprised of the positive vorticity difference located over Iran and Turkmenistan and negative difference located over northern India, Pakistan and southern China, indicates that the upper and lower vorticity fields are vertically collocated during all months, reflecting an equivalent barotropic structure (Fig. 8e-h). In-between these two vorticity anomalies is strong forcing for ascent (Martin 2006 and references therein) and thus the location of WCB formation and precipitation. The equivalent barotropic structure contrasts the vertical variation of the vorticity differences observed farther to the east over southern China indicative of a baroclinic structure. The upper-level cyclonic vorticity anomaly that is key in driving ascent and forming WCBs over Afghanistan is strongest between 350-200 hPa in all months and extends lower into the troposphere in January and February than March and April, despite strengthening in amplitude during the latter months. Conversely, the downstream

negative, anticyclonic vorticity difference remains of similar strength in all months. Considering how the streamfunction fields in Fig. 7 are the aggregate effect of all of the features in the vorticity field, it becomes clear why the cyclonic streamfunction differences are also stronger in April than in January. It also appears that the circulation associated with the two cyclonic vorticity features overwhelms the anticyclonic vorticity feature in-between (Fig. 8a-d), as the streamfunction fields mainly show the cyclonic differences (Fig. 7e-h). Composite isentropes on wet days tilt downwards towards the surface moving from the cyclonic to anticyclonic vorticity differences, reflecting warmer temperatures beneath the anticyclonic feature and cooler temperatures beneath the cyclonic feature.'

[Figure]

**Figure 8: Panels a) – d) the color shading shows the 200-350 hPa mean relative vorticity, ζ, difference, wet – dry days, for a) January, b) February, c) March, d) April, and the thick black lines show the location of the cross sections in panels e) – h). The color shading in panels e) – h) show the mean ζ difference, wet – dry days, and the black contours show the composite potential temperature, θ, on wet days. Units of ζ are 10⁻⁵ s⁻¹, and units of θ are K, contoured every 10 K. The think dashed black line is the location of topography along the cross section.**